# Cell division orientation is coupled to cell–cell adhesion by the E-cadherin/LGN complex

Martijn Gloerich[1,†], Julie M. Bianchini[1], Kathleen A. Siemers[1], Daniel J. Cohen[1] & W. James Nelson[1,2]

Both cell–cell adhesion and oriented cell division play prominent roles in establishing tissue architecture, but it is unclear how they might be coordinated. Here, we demonstrate that the cell–cell adhesion protein E-cadherin functions as an instructive cue for cell division orientation. This is mediated by the evolutionarily conserved LGN/NuMA complex, which regulates cortical attachments of astral spindle microtubules. We show that LGN, which adopts a three-dimensional structure similar to cadherin-bound catenins, binds directly to the E-cadherin cytosolic tail and thereby localizes at cell–cell adhesions. On mitotic entry, NuMA is released from the nucleus and competes LGN from E-cadherin to locally form the LGN/NuMA complex. This mediates the stabilization of cortical associations of astral microtubules at cell–cell adhesions to orient the mitotic spindle. Our results show how E-cadherin instructs the assembly of the LGN/NuMA complex at cell–cell contacts, and define a mechanism that couples cell division orientation to intercellular adhesion.

[1] Department of Biology, Stanford University, Stanford, California 94305, USA. [2] Department of Molecular and Cellular Physiology, Stanford University, Stanford, California 94305, USA. † Present address: Molecular Cancer Research, Center for Molecular Medicine, University Medical Center Utrecht, Universiteitsweg 100, 3584CG Utrecht, The Netherlands. Correspondence and requests for materials should be addressed to M.G. (email: m.gloerich@umcutrecht.nl) or to W.J.N. (email: wjnelson@stanford.edu).

The orientation of cell division defines the position of daughter cells within a tissue, and thereby controls tissue architecture and cell fate[1,2]. In simple epithelia, planar cell divisions maintain a single-layered epithelium[1,3], whereas divisions in the direction of the apico-basal axis induce multi-layering such as in stratified epithelia[2,4]. The importance of correct division orientation is underlined by various developmental disorders that are a consequence of misoriented cell division[5,6], which may also contribute to tumour progression[7–10].

The plane of cell division is specified by the position of the mitotic spindle. In tissues throughout the Metazoa this involves an evolutionarily conserved adaptor protein LGN that binds lipid-anchored $G\alpha_i$ at the cell cortex[11,12]. LGN localizes NuMA, which orients the mitotic spindle by anchoring spindle astral microtubules to the cell cortex and applying a pulling force on those microtubules through associated dynein[11,13–16]. To establish the correct orientation of the mitotic spindle, cells respond to instructive spatial cues from their local environment[17,18]. Although several cortical-binding sites for LGN have been described, including DLG[9,19], inscuteable[20–22] and afadin[23], the identities of the receptor(s) that sense and translate extracellular cues to localize the LGN/NuMA complex and thereby the mitotic spindle are not well understood.

In most tissues, neighbouring cells are coupled by evolutionarily conserved classical cadherins, such as E-cadherin. The cytosolic tail of E-cadherin is linked to the actin cytoskeleton through bound catenin proteins (α-, β- and p120-catenin), and forms a signalling platform that triggers intracellular responses following the *trans* engagement of the cadherin extracellular domain[24]. Importantly, loss of E-cadherin disrupts not only cell–cell adhesion but also the orientation of cell divisions, including the planar orientation of cell divisions in simple epithelia[25–29]. However, the precise role of E-cadherin in division orientation is not known, and it remains unclear whether E-cadherin merely plays a permissive role in division orientation or if E-cadherin itself is linked to the mitotic spindle[17]. Here, we demonstrate that LGN binds directly to the E-cadherin cytosolic tail, which directs the mitotic recruitment of NuMA, resulting in stable cortical associations of astral microtubules at cell–cell contacts to orient the mitotic spindle. In this way, E-cadherin directly coordinates two fundamental processes, cell–cell adhesion and cell division orientation, which control the organization of tissues during development and homoeostasis.

## Results

**E-cadherin recruits LGN to cell–cell contacts.** The polarized, cortical distribution of LGN defines the mitotic spindle axis in tissues throughout the Metazoa. However, it is not well understood how extracellular cues control LGN localization to direct spindle orientation. In MDCK epithelial cell monolayers, LGN was enriched at cell–cell contacts, whereas it was absent from membranes that were not in contact with neighbouring cells (Fig. 1a, top panels). This distribution of LGN at cell–cell contacts was even more pronounced after cells had entered mitosis (Fig. 1a, bottom panels). The specificity of LGN staining was confirmed by shRNA-mediated depletion, which resulted in a loss of LGN staining at cell–cell contacts (Supplementary Fig. 1).

Although multiple adhesion complexes contribute to intercellular adhesion in epithelial cells, E-cadherin plays a prominent role in establishing cell–cell adhesions[30]. We used a reductionist approach to test whether E-cadherin-based cell–cell adhesion directs the cortical enrichment of LGN. Single MDCK cells were plated on a surface functionalized with alternating stripes of collagen-IV and either the extracellular domain of E-cadherin (E-cad:Fc) or Fc as a control[31]. Total internal reflection fluorescence (TIRF) microscopy revealed significant LGN enrichment at the cell surface bound to E-cad:Fc stripes, compared with either collagen-IV or Fc (Fig. 1b). Thus, E-cadherin appears to direct the localization of this key component of the spindle orientation machinery to the cell cortex.

We considered how E-cadherin might establish the localization of LGN at cell–cell contacts. E-cadherin associates with the actin cytoskeleton through its cytosolic tail, which binds directly to p120-catenin and β-catenin. Despite little overlap in amino acid sequence, the TPR repeat domain of LGN (LGN–TPR)[20–22], and the armadillo repeat domain of β-catenin and p120-catenin[32,33] form a similar super-helical structure containing a charged groove (Fig. 1c; Supplementary Fig. 2). Moreover, binding of LGN–TPR to a negatively charged motif within NuMA is highly reminiscent of how β-catenin and p120-catenin bind to the E-cadherin cytosolic tail (Supplementary Fig. 2).

To test whether LGN–TPR binds directly to E-cadherin cytosolic tail, we co-expressed LGN–TPR and GST-tagged E-cadherin cytosolic tail in bacteria. A GST-pull down demonstrated that a complex formed between these two proteins (Fig. 1d). To test this interaction in mammalian cells, we mis-targeted E-cadherin cytosolic tail to the surface of mitochondria in U2OS cells that lack endogenous E-cadherin. This resulted in the co-recruitment of LGN–TPR (Fig. 1e). A similar co-recruitment of LGN occurred when the juxtamembrane domain (JMD), but not the β-catenin-binding domain (CBD) of E-cadherin cytosolic tail was targeted to mitochondria (Fig. 1e). LGN recruitment to E-cadherin cytosolic tail required a highly conserved, negatively charged motif within the JMD, as expression of a DEE758–760 triple alanine mutant (JMD-AAA) did not recruit LGN–TPR to mitochondria (Fig. 1e). Thus, E-cadherin cytosolic tail binds directly to LGN through a binding mode similar to that of cadherin-bound catenin proteins.

**E-cadherin instructs mitotic spindle orientation.** Previously, it was shown that loss of E-cadherin-mediated adhesion disrupts the orientation of cell division in different tissues, including planar division in epithelia[25–29]. Taken together with our identification of direct binding of E-cadherin to LGN, these results imply that E-cadherin could act as an instructive cue for mitotic spindle orientation that is sufficient to orient epithelial division. To test this directly, we selectively disrupted, induced or mimicked E-cadherin-mediated cell adhesion and analysed the localization of LGN and orientation of the mitotic spindle.

First, to disrupt E-cadherin adhesion without losing overall intercellular adhesion, we used MDCK cells expressing an E-cadherin mutant (T151) with a truncated, nonfunctional extracellular domain, but an intact plasma membrane-tethered cytosolic tail that binds catenins and actin (Supplementary Fig. 3a; ref. 34). Importantly, endogenous E-cadherin is down-regulated when T151 expression is induced, but a cohesive monolayer is maintained due to other cell–cell junctions (Supplementary Fig. 3a; ref. 34). Because it cannot form extracellular interactions, T151 E-cadherin localized along the entire plasma membrane, and LGN was similarly no longer enriched on the plasma membrane specifically at cell–cell contacts (Fig. 2a). Confocal imaging of centrosome-localized γ-tubulin showed that the planar spindle orientation observed in the presence of endogenous E-cadherin (Fig. 2b,c, +DOX) was also lost when E-cadherin was replaced by T151 (Fig. 2b,c, −DOX). This loss of spindle orientation in the presence of

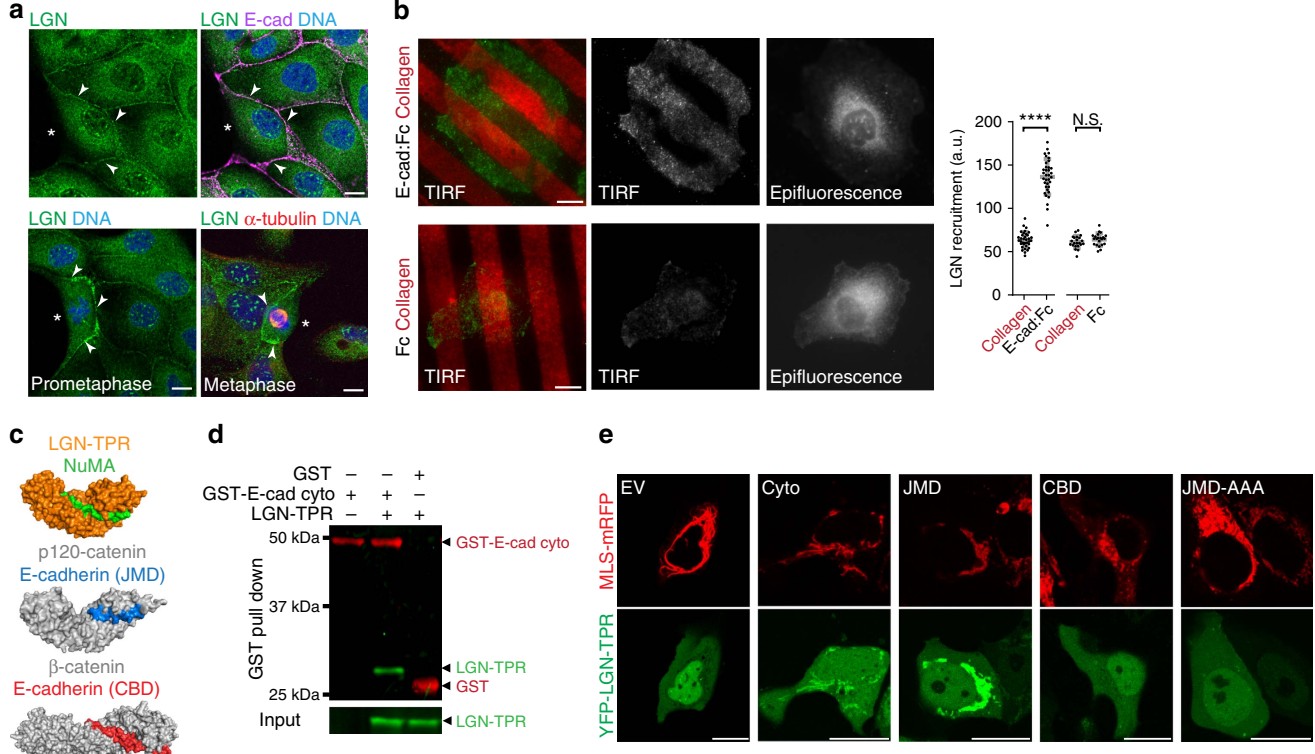

**Figure 1 | LGN is recruited to cell–cell contacts directly by E-cadherin. (a)** Localization of endogenous LGN at cell–cell contacts, marked with E-cadherin (E-cad), in interphase and mitotic MDCK cells. Arrowheads identify cell–cell contacts, and asterisks mark plasma membrane not contacting another cell. **(b)** TIRF and epifluorescence microscopy imaging of endogenous LGN in MDCK cells plated on surfaces micro-patterned with alternating stripes of collagen-IV/E-cad:Fc or collagen-IV/Fc, with a quantification of LGN staining intensities at the plasma membrane bound to different stripes. Quantified data were pooled from three independent experiments, grey bars show mean ± s.d. **(c)** Surface structures of NuMA in complex with the TPR repeats of LGN[20], and of β-catenin and p120-catenin with E-cadherin[32,33]. For more details of the E-cadherin/p120-catenin and LGN/NuMA-binding interface, see Supplementary Fig. 2. CBD, catenin binding domain; JMD, juxtamembrane domain. **(d)** GST-pull down of recombinant E-cadherin cytosolic tail (GST-E-cad-cyto) with LGN–TPR, immunoblotted for GST (red) and LGN (green). **(e)** U2OS cells, with low levels of endogenous E-cadherin, showing recruitment of YFP-tagged LGN–TPR to E-cadherin cytosolic tail (cyto) or JMD targeted to mitochondria by fusion with the mitochondrial localization signal (MLS) of ActA. This recruitment was not observed with CBD or the JMD containing a DEE758-760AAA mutation. EV, empty vector. Scale bars, 10 μm; ****$P < 0.0001$; N.S = not significant, using paired student $t$-test.

T151 E-cadherin was not limited to metaphase but extended to anaphase (Supplementary Fig. 4a,b); overexpression of the E-cadherin cytosolic tail has a similar effect[25]. Thus, selective loss of E-cadherin-mediated cell–cell adhesion disrupts the orientation of epithelial cell division in an otherwise cohesive monolayer that expresses the full repertoire of other cell–cell adhesion proteins.

Second, to test whether ectopic expression of E-cadherin could orient the mitotic spindle, we used mouse L-cell fibroblasts that lack E-cadherin and other cell–cell adhesion junctions. Ectopic expression of E-cadherin leads to the up-regulation of the cadherin-associated catenins and formation of E-cadherin-based cell–cell contacts in these cells (Supplementary Fig. 3b; refs 35,36). Expression of E-cadherin induced the recruitment of endogenous LGN to cell–cell contacts (Fig. 2d), and a significant bias in spindle orientation parallel to the formed monolayer (Fig. 2e,f). In parental L-cells, in contrast, LGN was not enriched at the plasma membrane and spindle orientation was random (Fig. 2e,f). Ectopic expression of DE-cadherin in non-adherent *Drosophila* S2 cells, grown in suspension and thus also in the absence of extracellular matrix adhesions, also biased spindle orientation perpendicular to cell–cell junctions (Supplementary Fig. 3c–e). This was not observed when cell–cell adhesion was induced in S2 cells by expression of either a DE-cadherin mutant lacking the cytosolic tail, or the unrelated cell–cell adhesion molecule Fasciclin II (Supplementary Fig. 3d,e).

Third, to test whether E-cadherin is sufficient to instruct spindle orientation in epithelial cells, we used micro-fabricated silicone sidewalls functionalized with E-cad:Fc. These sidewalls have a geometry similar to native E-cadherin adhesions in MDCK cells[37], and mimic E-cadherin-mediated cell–cell junctions in the absence of other adhesion proteins and juxtamembrane signals from neighbouring cells (Fig. 3a). MDCK cells adhered to E-cad:Fc-coated sidewalls with strong enrichment of cellular E-cadherin at the plasma membrane at the cell-sidewall contact (Fig. 3b). Significantly, MDCK cell attachment to an Ecad:Fc sidewall resulted in the recruitment of LGN to levels comparable to native cell–cell junctions (Fig. 3c), and biased the orientation of the mitotic spindle perpendicular to the sidewall (Fig. 3d,e; Supplementary Fig. 4c). In contrast, spindle orientation was random in cells that associated with control sidewalls coated with Fc (Fig. 3d,e; Supplementary Fig. 4c). Altogether, our results demonstrate that E-cadherin is sufficient to act as an instructive cue that recruits LGN and orients the mitotic spindle relative to cell–cell junctions.

**E-cadherin instructs spindle orientation through bound LGN.**
To unambiguously demonstrate that E-cadherin instructs

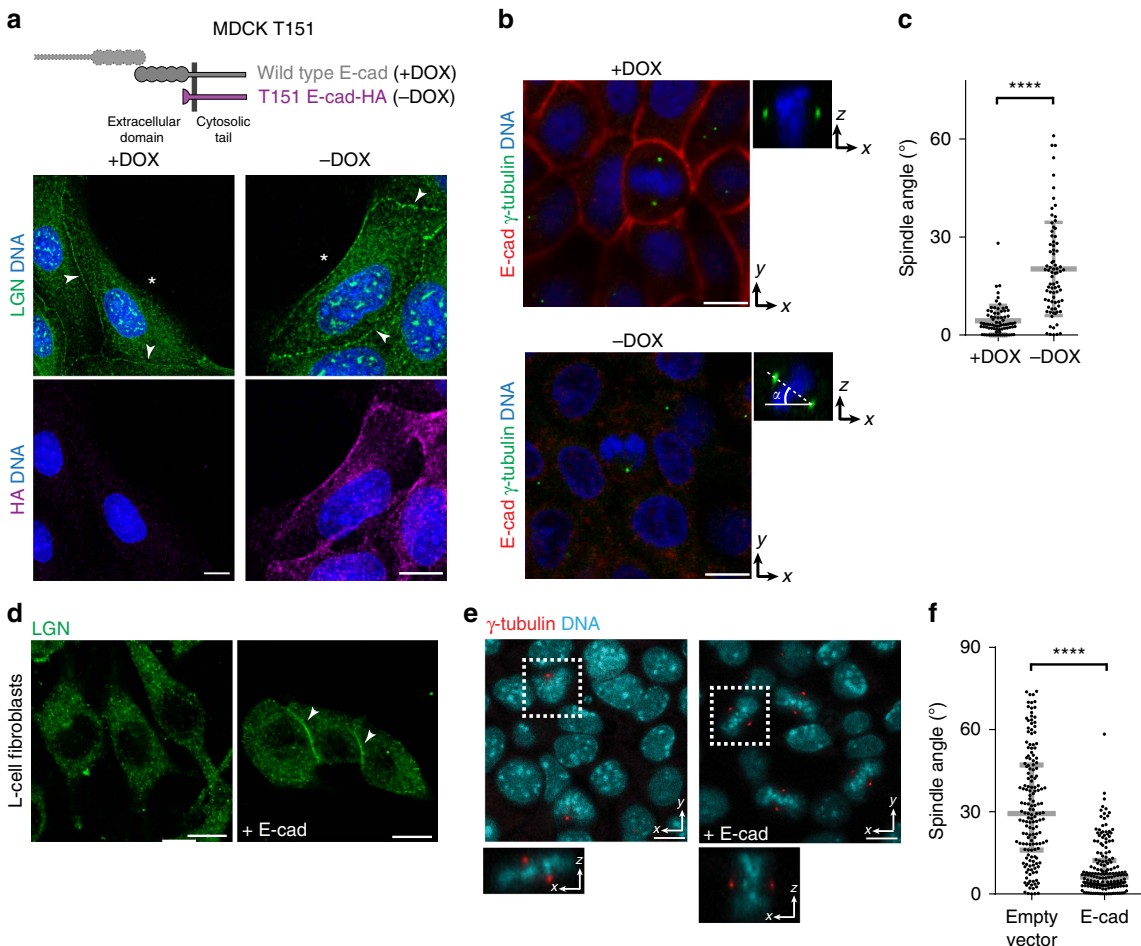

**Figure 2 | E-cadherin is an instructive cue for mitotic spindle orientation. (a)** Schematic representations of full-length and T151 truncated E-cadherin (under control of the doxycycline (DOX) repressible promotor). Immunostaining of endogenous LGN and the HA-tagged T151 E-cadherin ( − DOX) showing that the plasma membrane localization of LGN is no longer restricted to cell–cell contacts (arrowheads) but also localizes at membranes not in contact with neighbouring cells as well (asterisks). **(b)** Confocal images of γ-tubulin labelled centrosomes in MDCK T151 cells showing the orientation of the mitotic spindle in metaphase. **(c)** Quantification of mitotic spindle angle relative to the basal surface in MDCK T151 cells. **(d)** Localization of LGN in L-cells in the absence or presence of ectopic E-cadherin (E-cad). **(e)** Confocal images of γ-tubulin labelled centrosomes in L-cells, showing the orientation of the mitotic spindle in metaphase, in the absence or presence of ectopic E-cadherin. **(f)** Quantification of mitotic spindle angle relative to the basal surface in L-cells in the absence (empty vector) or presence of ectopic E-cadherin. Scale bars, 10 μm. Quantified data were pooled from three independent experiments. Grey bars in dot plots show mean ± s.d.; \*\*\*\*$P < 0.0001$ using Mann–Whitney test.

cell division orientation through formation of a complex with LGN, we designed an E-cadherin mutant that does not bind LGN but retains its ability to establish cell–cell adhesion. We showed that LGN binds to a conserved, negatively charged amino acid triplet (DEE758–760) within the JMD of E-cadherin cytosolic tail (Fig. 1e). This amino acid triplet is also essential for p120-catenin binding to E-cadherin, which requires a second negatively charged triplet EED764-766 (Fig. 4a; ref. 33). We tested the contribution of individual residues within the first triplet for binding p120-catenin or LGN using the mitochondria co-recruitment assay. A D758A mutant selectively blocked the recruitment of LGN–TPR to E-cadherin cytosolic tail, similar to the effect of mutating the entire triplet (Fig. 4b,c). D758A, however, did not affect p120-catenin recruitment in this assay (Fig. 4d), presumably because the adjacent charged residues provide sufficient affinity for p120-catenin binding (Fig. 4a). Significantly, a full-length E-cadherin D758A mutant expressed in L-cells localized at cell–cell junctions and induced cell–cell adhesion comparable to wild-type E-cadherin (Fig. 4e). However, E-cadherin D758A did not induce planar spindle orientation

(Fig. 4e). Similarly, planar spindle orientation was blocked in L-cells-expressing wild-type E-cadherin but depleted of LGN by shRNA-mediated knockdown (Fig. 4e). Thus, the localization of LGN is regulated by its interaction with the E-cadherin cytosolic tail, and formation of this complex is required for E-cadherin-mediated spindle orientation.

**E-cadherin localizes the LGN/NuMA complex in mitosis.** LGN orients the mitotic spindle by localizing its binding partner NuMA to the cortex[11]. However, the similarity in binding mode between LGN to E-cadherin and to NuMA (Fig. 1c; Supplementary Fig. 2) implies that these interactions are likely mutually exclusive. We tested this possibility in an *in vitro* competition assay, which showed that addition of the recombinant LGN-binding domain of NuMA competed E-cadherin from the E-cad-cyto/LGN–TPR complex (Fig. 5a). This raises the question whether E-cadherin is able to localize NuMA, even though a ternary complex of E-cadherin, LGN and NuMA cannot form.

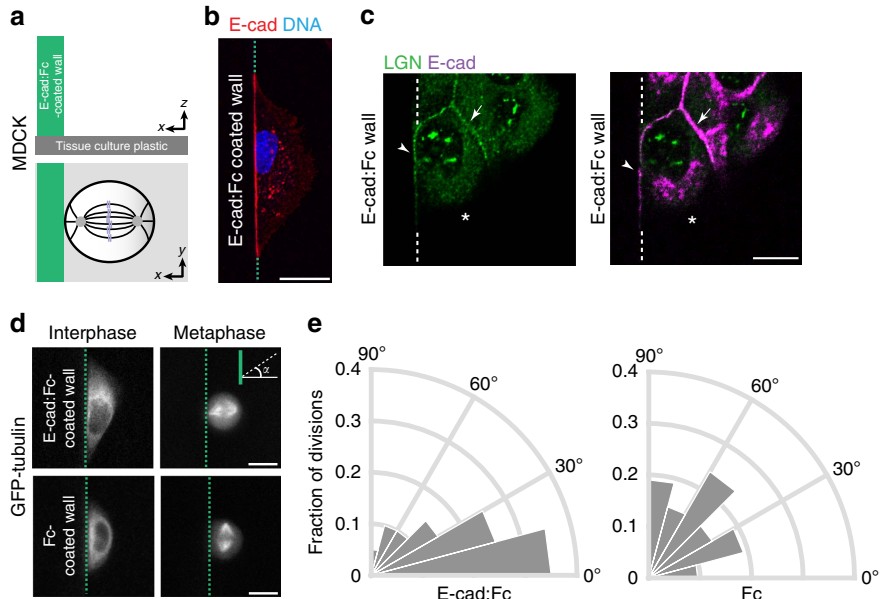

**Figure 3 | An artificial E-cadherin cell–cell junction orients epithelial cell divisions.** (**a**) Design of the E-cadherin:Fc sidewall in which a vertical silicone sidewall is coated with E-cadherin:Fc (E-cad:Fc) and the basal surface is uncoated. (**b**) Localization of E-cadherin:DsRed (E-cad) at the sidewall-cell junction in MDCK cells. (**c**) Immunostaining of endogenous LGN in cells associated with an E-cadherin:Fc sidewall, showing that LGN is recruited to the sidewall-cell junction marked with E-cadherin (arrowhead), similar to native cell–cell adhesions (arrow) in the monolayer, whereas LGN is absent from the free membrane (asterisk). (**d,e**) GFP-tubulin expressing MDCK cells associated with E-cad:Fc ($n = 20$) and Fc control ($n = 21$) sidewalls, with rose diagram quantification of the mitotic spindle angle relative to the sidewall. See Supplementary Fig. 3f for unbinned data. Scale bars, 10 μm. Quantified data were pooled from four independent experiments.

NuMA contains a nuclear localization motif and, therefore, in contrast to LGN resides primarily in the nucleus during interphase (Fig. 5b). Following nuclear envelope breakdown at the onset of mitosis, NuMA redistributed to the mitotic spindle and cell cortex (Fig. 5b). TIRF imaging of cells plated on collagen-IV/E-cad:Fc micro-patterns showed that NuMA accumulated at sites of E-cadherin adhesion in mitotic cells (Fig. 5c,d). To test whether loss of nuclear retention on nuclear envelope breakdown was the key event in regulating NuMA targeting to E-cadherin adhesions in mitosis, we deliberately over-expressed GFP-NuMA to saturate its nuclear presence and cause its cytoplasmic accumulation in interphase (Fig. 5e). Under these conditions, cytosolic GFP-NuMA also co-localized with sites of E-cadherin adhesion on micro-patterned surfaces (Fig. 5f). This result implies that localization of NuMA to E-cadherin adhesions is a constitutive process, but it occurs specifically in mitosis when NuMA is released from the nucleus.

Since a ternary complex of E-cadherin, LGN and NuMA does not form, we propose a parsimonious explanation for the recruitment of NuMA to E-cadherin. Following junctional enrichment of LGN by E-cadherin in interphase, the mitotic release of NuMA from the nucleus competes LGN from E-cadherin, as we demonstrated in vitro (Fig. 5a); the resulting LGN/NuMA complex remains localized at E-cadherin-based cell–cell contacts by LGN binding to cortical $G\alpha_i$ (ref. 11). To test this hypothesis, we examined whether the recruitment of NuMA to E-cadherin adhesions was dependent on both LGN and the interaction of LGN with $G\alpha_i$. As we hypothesized, NuMA was not recruited to E-cadherin adhesions in mitosis after shRNA-mediated depletion of LGN (Fig. 5g; Supplementary Fig. 5a,b). Because LGN binding depends on the GDP-bound state of $G\alpha_i$ (ref. 11), we next over-expressed the guanine nucleotide exchange factor Ric8 to drive $G\alpha_i$ into the GTP-bound state and thereby disrupt binding to LGN, as previously demonstrated[38].

Similar to depletion of LGN, this resulted in loss of NuMA recruitment to micro-patterned E-cadherin adhesions in mitotic cells (Fig. 5g; Supplementary Fig. 5c). Importantly, Ric8 overexpression did not affect the junctional localization of E-cadherin itself (Supplementary Fig. 5d). Finally, siRNA-mediated depletion of $G\alpha_i$ from MDCK cells resulted in a similar loss of NuMA recruitment to micro-patterned E-cadherin adhesions in mitosis (Fig. 5g). These results indicate that E-cadherin serves as a cortical landmark to localize LGN to cell–cell adhesions, which primes the site for mitotic assembly of the LGN/NuMA complex bound to $G\alpha_i$ at cell–cell adhesions (Fig. 5h).

**E-cadherin stabilizes cortical astral microtubule associations.** NuMA provides the link between LGN and the mitotic spindle, as NuMA binds directly to astral microtubules and is able to apply a pulling force on those microtubules through its association with dynein[11,13–16]. Having established that the E-cadherin/LGN complex directs the mitotic localization of NuMA to cell–cell adhesions (Fig. 5), we hypothesized that mitotic microtubules should form more stable associations with the cell cortex at E-cadherin-dependent cell–cell adhesions than at non-adhesion sites. To test this, MDCK cells stably expressing mRFP-tagged EB1, which marked the plus ends of microtubules, were plated onto E-cad:Fc functionalized glass to induce E-cadherin-dependent adhesion on the basal surface of cells. Alternatively, cells were plated onto a control surface coated with poly-D-lysine to induce nonspecific, electrostatic adhesion. Cortical EB1-labelled microtubules in cells in metaphase were visualized by TIRF microscopy (Fig. 6a). EB1-marked microtubules in cells adhering to poly-D-lysine had short-lived associations with the basal membrane (Fig. 6b,c; average dwell time ∼7 s). By comparison, cortical EB1-plasma membrane

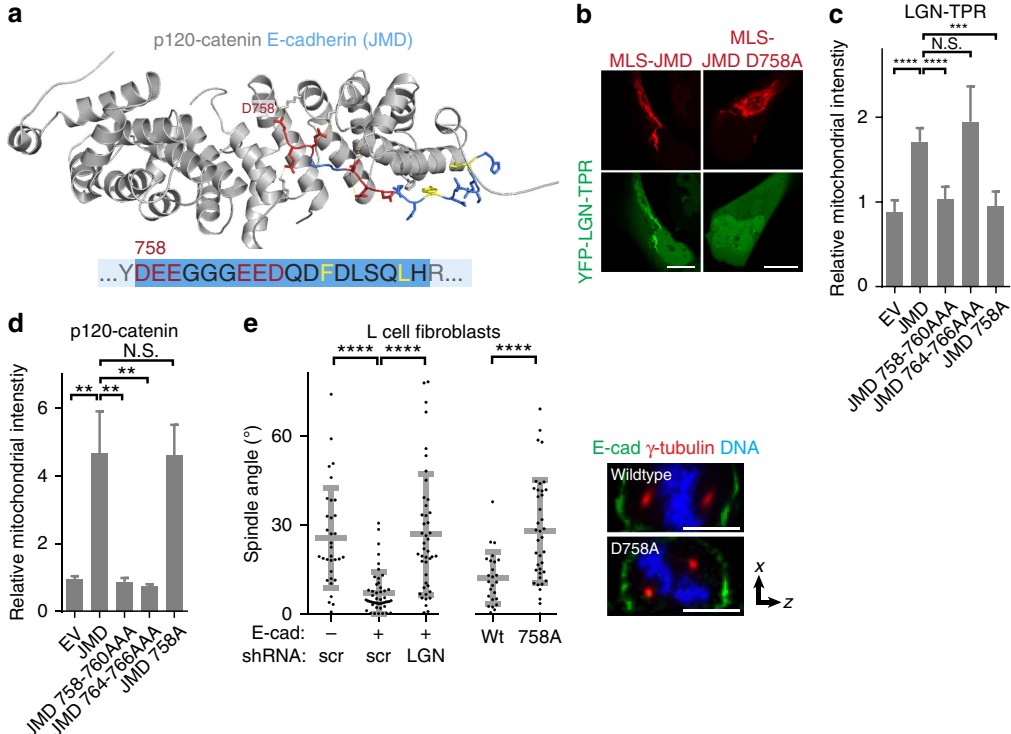

**Figure 4 | E-cadherin-induced spindle orientation is dependent on LGN binding.** (**a**) Three-dimensional structure of the interaction between p120-catenin and the juxtamembrane domain (JMD) of E-cadherin, indicating the two negatively charged triplets (red) and hydrophobic residues (yellow) within E-cadherin that contribute to p120-catenin binding; based on ref. 33. (**b,c**) Recruitment of YFP–LGN–TPR to wild-type or D758A JMD, which was targeted to mitochondria by fusion with the mitochondrial localization signal (MLS) of ActA. The quantification shows the ratio of mitochondrial versus cytosolic YFP–LGN–TRP on co-expression of empty vector (EV), wild-type JMD or the indicated mutants; $n = 5$. (**d**) Quantification of the recruitment of GFP-p120-catenin to JMD, containing indicated point mutations, and targeted to mitochondria by fusion with the MLS of ActA; $n = 5$. (**e**) Quantification of mitotic spindle angle relative to the basal surface of parental L-cells, L-cells-expressing E-cadherin with either scrambled (scr) or LGN shRNAs, or L-cells-expressing wild-type or D758A E-cadherin-GFP, pooled from three independent experiments. Right: $xz$-projection of mitotic cell showing similar lateral distribution at cell–cell contacts of wild-type and D758A E-cadherin-GFP. Quantified data were pooled from three independent experiments, grey bars in dot plots show mean ± s.d. Scale bars, 5 μm. **$P < 0.003$, ***$P < 0.0002$, ****$P < 0.0001$, N.S. = not significant using unpaired student $t$-test (**c,d**) or Mann–Whitney test (**e**).

dwell times in cells adhering to E-cad:Fc were significantly longer (Fig. 6b,c; average dwell time ∼25 s), although the total number of events per minute was comparable to cells on poly-D-lysine. Significantly, dwell times of EB1-plasma membrane events on E-cad:Fc associated membranes were reduced on shRNA-mediated depletion of LGN (Fig. 6d). Thus, the cortical association of mitotic astral microtubules is stabilized at sites of E-cadherin adhesion, in an LGN-dependent manner.

## Discussion

Both oriented cell division and cell–cell adhesion play prominent roles in tissue development and organization. Here, we provided evidence that E-cadherin coordinates these two fundamental processes, as we demonstrated that E-cadherin is an instructive cue for mitotic spindle orientation. Moreover, we identified the underlying molecular mechanism by which E-cadherin orients the mitotic spindle, by mediating the successive, cell-cycle dependent recruitment of LGN and NuMA to cell–cell contacts. This directs the formation of stable cortical attachments of mitotic microtubules to E-cadherin-based adhesions, and the alignment of the mitotic spindle with cell–cell contacts. In this way, E-cadherin serves as a spatial cellular landmark that couples cell division orientation directly to cell–cell adhesion.

Our data indicate that LGN binds the cytosolic tail of E-cadherin in a manner similar to that of the cadherin-associated catenins. First, despite little overlap in amino acid sequence, the TPR repeat domain of LGN adopts a super-helical structure containing a charged groove similar to the armadillo repeat domains of p120-catenin and β-catenin (Supplementary Fig. 2), and binds directly to the E-cadherin cytosolic tail (Fig. 1). Second, mutational analyses showed that LGN binding required a negatively charged region within the E-cadherin JMD, which is reminiscent of how p120- and β-catenin bind to E-cadherin, and LGN binds to NuMA (Fig. 1e). Finally, binding of p120-catenin and LGN to the cytosolic tail of E-cadherin is mutually exclusive (Supplementary Fig. 6). Altogether, these findings imply a comparable mode of binding of LGN and p120-catenin to an overlapping region within E-cadherin. Using a mutation within E-cadherin that selectively disrupts LGN recruitment but leaves its association with p120-catenin intact, we demonstrated that its association with LGN is essential for E-cadherin-induced mitotic spindle orientation (Fig. 4).

Planar epithelial divisions have been attributed to other cellular factors, including: (1) cell polarization in the apico-basal axis, which is established downstream of E-cadherin adhesion[39,40]; (2) laterally localized proteins that are indirectly targeted to cell–cell contacts by E-cadherin (for example, DLG[9,41] and APC[25,29]); and (3) other cell–cell adhesion complexes that are formed following E-cadherin adhesion (for example, JAM-A[42]). Although these factors imply that E-cadherin may indirectly influence

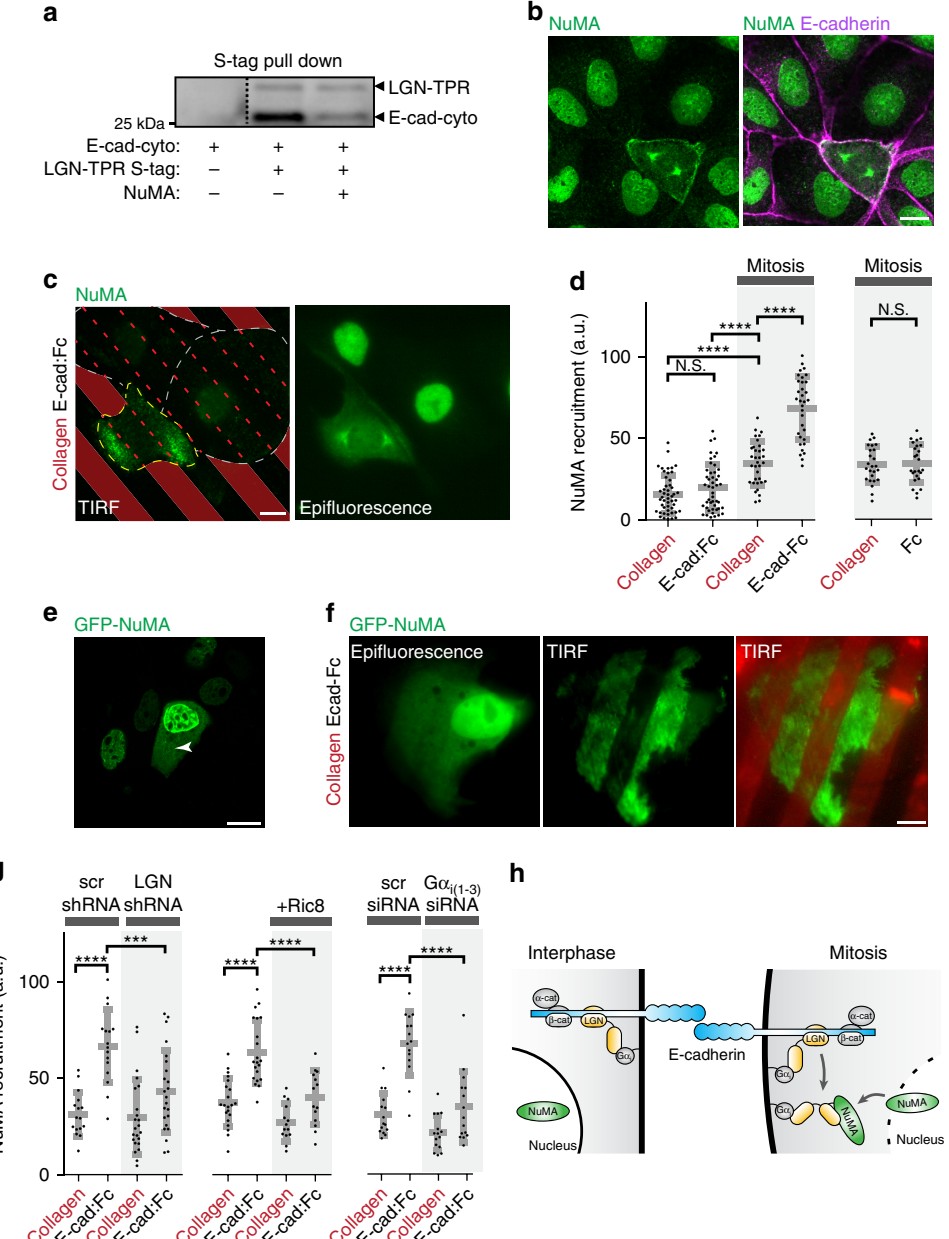

**Figure 5 | E-cadherin localizes the mitotic LGN/NuMA complex to cell–cell contacts.** (**a**) S-protein agarose pull down of recombinant LGN–TPR and E-cadherin cytoplasmic tail in the presence or absence of NuMA peptide, immunoblotted for LGN and E-cadherin, showing the competition of E-cadherin cytosolic tail from LGN–TPR:S-tag by the LGN-binding domain of NuMA. (**b**) Localization of endogenous NuMA and E-cadherin in interphase and mitotic MDCK cells. (**c**) TIRF and epifluorescence microscopy of NuMA in MDCK cells plated on micro-patterned alternating stripes of collagen-IV/E-cad:Fc. Cells are outlined in yellow (mitotis) and grey (interphase), for increased clearness stripes are shown as projections in the TIRF image. (**d**) Quantification of NuMA fluorescence intensity at the plasma membrane bound to stripes coated with different proteins in interphase and mitotic cells. (**e**) Cytosolic localization of over-expressed GFP-NuMA in interphase MDCK cells, showing its cytosolic accumulation (marked with an asterisk) in cells with high expression. (**f**) Localization of GFP-NuMA on MDCK cells plated on collagen-IV/E-cad:Fc micro-patterned surfaces visualized by TIRF microscopy. (**g**) Quantification of NuMA fluorescence intensity at the plasma membrane bound to different stripes in mitotic cells on shRNA-mediated depletion of LGN, disruption of LGN/Gα$_i$ interactions by overexpression of Ric8, or on expression of siRNAs targeting Gα$_i$1, Gα$_i$2 and Gα$_i$3. Representative images and epifluorescence of NuMA localization in LGN shRNA monolayers are shown in Supplementary Fig. 4a–4c. (**h**) Model for localization of the LGN/NuMA complex to cell–cell contacts by E-cadherin. By interacting with the cytosolic tail of E-cadherin, LGN is recruited to cell–cell contacts in interphase. After nuclear breakdown during mitotic entry, NuMA is released from the nucleus and competes LGN from the E-cadherin/LGN complex. The locally formed LGN/NuMA complex remains at E-cadherin adhesions through the interaction of LGN with Gα$_i$. Scale bars, 10 µm; all quantified data were pooled from three independent experiments; grey bars in dot plots show mean ± s.d.; ****$P < 0.0001$; ***$P = 0.0002$, N.S. = not significant, using unpaired student $t$-test.

epithelial cell division orientation, we showed that E-cadherin in fact plays a direct role. Our reductionist approaches provided evidence that E-cadherin is sufficient to orient epithelial divisions in the absence of other junctional complexes or juxtamembrane signalling (Figs 2 and 3), and that direct binding between E-cadherin and LGN is essential for this (Fig. 4). Since E-cadherin

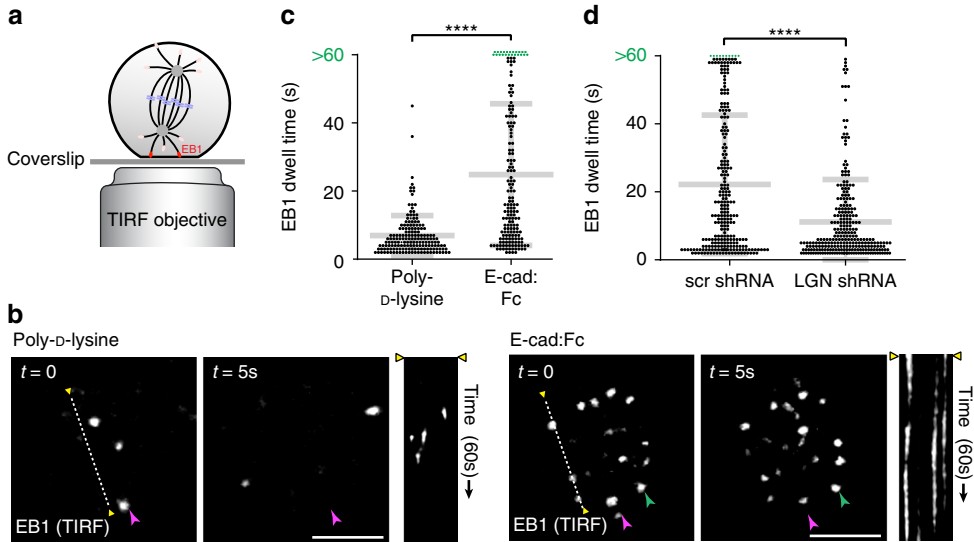

**Figure 6 | E-cadherin adhesions stabilize cortical astral microtubule associations.** (**a,b**) TIRF microscopy imaging of mRFP-EB1-marked astral microtubules at the basal membrane of MDCK cells adhering to poly-D-lysine or E-cad:Fc-coated surfaces. The kymograph shows the duration of basal mRFP-EB1 events in a 60 s period along the indicated dotted line. Purple and green arrowheads indicate dynamic (<5 s duration) and stable (>5 s duration) EB1 puncta, respectively. (**c**) Quantification of dwell times of EB1 puncta at the basal membrane of mitotic MDCK cells adhering to poly-D-lysine or E-cad:Fc-coated surfaces (*n* = 14 cells). (**d**) Quantification of the dwell times of EB1 puncta at the basal membrane of mitotic MDCK cells, expressing scrambled control (scr) or LGN shRNA, adhering to E-cad:Fc-coated surfaces (*n* = 11 cells). Scale bars, 5 μm. Quantified data were pooled from three independent experiments; grey bars show mean ± s.d. ****P < 0.0001 using Mann–Whitney test.

is a ubiquitous cell–cell adhesion protein, the E-cadherin/LGN complex may also be important in division orientation in non-epithelial cells, for instance in asymmetric divisions of *Drosophila* germ line stem cells that require local *Drosophila* E-cadherin adhesions[26,29].

We do not exclude the possibility that other pathways may impinge on the cortical recruitment of LGN/NuMA by E-cadherin. Our data imply that LGN is competed from E-cadherin on the release of NuMA from the nucleus during mitotic entry, and the formed LGN/NuMA complex remained at E-cadherin adhesions bound to $G\alpha_i$ (Fig. 5). Additional mechanisms could aid in restricting the LGN/NuMA complex to cell–cell contacts. For example, regulation of the association of LGN with $G\alpha_i$, through aPKC-mediated phosphorylation of LGN[39,40] or GDP/GTP exchange regulation $G\alpha_i$ by Ric8 (ref. 43) could limit the lifetime of LGN bound to the plasma membrane and thus restrict its lateral diffusion. Furthermore, while our data imply that the release of NuMA from the nucleus is what triggers the formation of the LGN/NuMA complex at E-cadherin adhesions (Fig. 5), mitotic post-translational modifications of E-cadherin[44], LGN or NuMA[18] may further regulate these interactions.

The existence of additional pathways for orienting the spindle further implies regulatory redundancy, which underscores the importance of planar division orientation in epithelial development and homoeostasis. These pathways may be particularly important in specialized epithelia in which E-cadherin localization is more restricted, such as in invertebrate epithelia where E-cadherin is enriched at the apex of the lateral membrane[45]. In addition, other extracellular cues may instruct cell division orientation through competition with E-cadherin to adjust the cortical localization of LGN/NuMA, for example through apically localized mInsc/Inscuteable[20–22]. These cues may help to adapt spindle orientation to specific tissue requirements, for instance to induce perpendicular divisions and thereby epithelial multi-layering[2]. It will be interesting to test how the balance between competing cues is regulated,

in particular in tissues in which LGN displays distinct localizations corresponding to differently oriented divisions[46].

## Methods

**Antibodies.** The following commercial antibodies were used at the indicated concentrations for western blot (WB) and immunofluorescence (IF): alpha-tubulin (DM1A, Sigma, 1:1,000 IF), DE-cadherin (DCAD2; T. Uemura, Kyoto University, Sakyo-ku, Kyoto, Japan; Developmental Studies Hybridoma Bank, 1:200 IF), DECMA-1 (GeneTex, GTX11512, 1:500 IF), Gamma-tubulin (Sigma, T6557 Clone GTU-88, 1:1,000 IF), GST (4C10, Covance, 1:500 WB), HA (Sigma, H6908, 1:250 IF), LGN (Millipore, ABT-174, 1:250 IF and Sigma, SAB2100965, 1:5,000 WB), and rr1 (mouse; Developmental Studies Hybridoma Bank, 1:250 IF). The E2 (rabbit, 1:250 IF) antibody recognizing the cytosolic tail of E-cadherin was generated in-house.

**Recombinant protein expression and binding experiments.** GST-tagged cytosolic tail of mouse E-cadherin (NM_009864.2, amino acids 736–783) and the TPR repeats of human LGN (NM_013296.4, amino acids 8–311) were cloned into the BamHI/NotI and EcoRV/KpnI restriction sites of the pET-Duet-1 dual expression plasmid (Novagen), respectively. Control plasmids contained either only GST-tagged cytosolic tail, or LGN–TPR and GST. Constructs were transformed into BL21 (DE3) *E. coli* and protein expression was induced with 1 mM IPTG for 6 h at 30 °C. Bacteria were subsequently lysed using an Emulsiflex B-15 (Avestin) in a buffer containing 20 mM Tris–HCl pH 8, 200 mM NaCl, 2 mM EDTA, 5 mM DTT and protease inhibitor cocktail V (Millipore). Lysates were cleared by high-speed centrifugation and E-cadherin/LGN–TPR complexes were isolated by coupling of 100 μl lysate to glutathione agarose beads (Agarose Beads Technologies) followed by extensive washing in a wash buffer containing 20 mM Tris–HCl pH 7.5, 150 mM NaCl, 2.5 mM DTT, 0.005% Tween, 2.5 % glycerol and protease inhibitor cocktail V (Millipore), all at 4 °C. Bound proteins were eluted in Laemmli buffer and analysed by SDS–PAGE and western blotting with GST and LGN antibodies.

The region of human NuMA containing the LGN-binding site (NM_006185, amino acid 1808–1988) was cloned into a pGEX-TEV plasmid using EcoRI and SacI restriction sites and expressed similarly as the pET-Duet-1 constructs. Protein was isolated on glutathione agarose beads, and the GST was cleaved off with tobacco etch virus (TEV) protease and purified on a MonoQ anion exchange column (GE Healthcare Life Sciences) in 20 mM Tris pH 8, 1 mM DTT, and a NaCl gradient from 0 to 1 M. For competition experiments, an S-tag was cloned C-terminally to LGN-TPR, and GST-E-cad-cyto/ LGN–TPR:S-tag complexes were isolated on glutathione agarose beads as described above. The GST-tag was cleaved off with TEV protease, and the eluate was subsequently coupled to S-protein agarose beads (Millipore) for 1 h to specifically

enrich for the E-cad-cyto/LGN–TPR complex in the absence of free E-cad-cyto. After washing the beads, 100 μg NuMA was added and incubated at 4 °C overnight to compete off LGN–TPR from E-cad-cyto. Beads were washed, and E-cad-cyto and LGN–TPR were eluted in Laemmli buffer and analysed by SDS–PAGE and western blotting.

The extracellular domain of E-cadherin fused to the Fc domain of human IgG1, E-cad:Fc, was purified by collecting culture media from HEK293 cells expressing pcDNA3-E-cad:Fc cultured in the absence of fetal bovine serum (FBS) for 2–3 days to minimize the presence of serum-derived IgGs during protein purification. Culture media was cleared by centrifugation, and E-cad:Fc was purified over a Protein A column (Pierce).

**Cell culture and transfections.** Parental Madin-Darby Canine Kidney (MDCK) GII cells[47], and MDCK cells stably expressing E-cadherin-DsRed[48], GFP-α-tubulin[49], mRFP-EB1 (kindly provided by A. Barth, Stanford University) or truncated E-cadherin (T151) under control of a doxycycline-repressible promotor[34] were cultured at 37 °C and 5% $CO_2$ in air in low glucose DMEM, 10% FBS, $1 \, g \, l^{-1}$ sodium bicarbonate and penicillin/streptomycin/kanamycin. U2OS, HEK293T, parental L-cells, L-cells-expressing murine E-cadherin under control of a dexamethasone-inducible promotor (LE-cells) and HEK293 E-cadherin:Fc cells were cultured in high glucose DMEM, 10% FBS, $1 \, g \, l^{-1}$ sodium bicarbonate and penicillin/streptomycin/kanamycin. All live-cell imaging was performed with the same media formulations in the absence of phenol red. *Drosophila* S2U cells expressing full-length DE-cadherin, DE-cadherinΔCyto or Fasciclin II cloned into pAc5.1/V5–His B were maintained in Schneider's medium (Invitrogen) supplemented with 10% FBS (Sigma-Aldrich), penicillin/streptomycin/kanamycin, $20 \, \mu g \, ml^{-1}$ blasticidin and $125 \, \mu g \, ml^{-1}$ hygromycin, as previously described[50].

MDCK GII cells were transfected using Lipofectamine 3000 (Life Technologies) and U2OS cells transfected using Xtremegene HP (Roche), according to the manufacturer's protocols. For lentiviral-knockdown experiments, cells were infected with lentiviral shRNAs targeting LGN (TRCN0000011025, Mission shRNA plasmid, Sigma-Aldrich) or scrambled control (SHC002, Mission shRNA plasmid, Sigma-Aldrich) produced in HEK293T cells, and analysed at least 3 days after infection. ON-TARGETplus siRNAs against Gnai1, Gnai2 and Gnai3 were obtained from Dharmacon, pEGFP-C1-NuMA was a gift from Michael Mancini (Addgene plasmid #81029).

**Spindle orientation measurements.** MDCK T151 cells were cultured in the absence or presence of doxycycline ($1 \, \mu g \, ml^{-1}$) and $1.5 \times 10^6$ cells were seeded onto glass-bottom imaging dishes (Mattek) pre-coated with rat tail collagen type I (Corning). LE-cells were cultured in the absence or presence of 1 μM dexamethasone (Sigma-Aldrich), and $1.5 \times 10^6$ cells were seeded onto glass-bottom imaging dishes pre-coated with poly-D-lysine (Sigma-Aldrich) to minimize matrix adhesions. The following day, cells were fixed in methanol, blocked in buffer containing 1% BSA, 1% goat serum (GS) and 1% donkey serum (DS), incubated with the indicated antibodies, and subsequently with Alexa-labelled secondary antibodies (Life Technologies) and Hoechst 33342 (Molecular Probes). Stained cells were analysed with a LSM 710 confocal microscope using a 63 × objective (N.A. 1.4) and the angle of the mitotic spindle in metaphase was measured in Z-stacks of cells of at least three independent experiments using ImageJ software (NIH). Data showed a non-Gaussian distribution and were statistically analysed using the Mann–Whitney test.

*Drosophila* S2 cells expressing DE-cadherin, DE-cadherinΔcyto or Fasciclin II were resuspended at $1 \times 10^6$ cells $ml^{-1}$ in T25 flasks (Corning) and gently swirled for 2 h at 250 r.p.m. to induce the formation of cell–cell contacts. Cells were subsequently fixed in 4% paraformaldehyde and transferred to coverslips pre-coated with poly-L-lysine (Sigma-Aldrich), followed by permeabilization in buffer containing 0.1 % Triton X100, incubation with blocking buffer containing 1% BSA, 1% GS and 1% DS, and incubation with the indicated antibodies and subsequently Alexa-labelled secondary antibodies (Life Technologies) and Hoechst 33342 (Molecular Probes). Cells were imaged with a Zeiss Axiovert 200 with a 63 × objective (N.A. 1.4), using an AxioCam mRM camera and AxioVision Rel. 4.6 software (Carl Zeiss MicroImaging). Pairs of contacting cells were identified, and the angle of the mitotic spindle in metaphase cells, relative to the cell–cell contacts, was measured using ImageJ software (NIH). Data showed a non-Gaussian distribution and was analysed statistically using the Mann–Whitney test.

**Functionalized sidewalls.** Sidewall scaffolds were constructed from 250 μm thick silicone sheeting (Bisco HT-6240, Stockwell Elastomers) cut into microwell inserts measuring 14 × 11 mm using a computer-controlled razor writer (Cameo, Silhouette). Stencils were plasma activated using atmospheric plasma (50 W, 45 s, 500 mTorr, PDC-001, Harrick Plasma) and subsequently silanized by immersion in a solution of 2% Triethoxysilylundecanal (Gelest) and 2% Triethylamine (Sigma) in pure ethanol (Goldshield), followed by baking at 85 °C for 3 h. A total of $250 \, \mu g \, ml^{-1}$ Protein A/G (Thermo-Pierce) in sodium cyanoborohydride coupling buffer (Sigma) was added to each well and allowed to react with the silanized stencils overnight at 4 °C. Stencils were washed with PBS-containing 0.1% Tween-20 (PBST), incubated for at least 1 h with

$200 \, \mu g \, ml^{-1}$ E-cadherin:Fc or human IgG Fc (Abcam) at room temperature, and then with $10 \, mg \, ml^{-1}$ BSA for 30 min to neutralize remaining aldehydes and block exposed silicone. Functionalized stencils were transferred to tissue culture plastic imaging dishes (Ibiditreat, Ibidi), and $2 \times 10^3$ MDCK cells stably expressing E-cadherin:DsRed or GFP-tubulin were seeded in each well, which were tilted during plating to promote interactions of cells with the sidewalls. Live E-cadherin:DsRed MDCK cells were imaged 3 h after plating in the presence of NucBlue (Molecular Probes) using a Leica SP8 scanning confocal microscope and a 63 × (1.4 N.A.) objective in a temperature controlled incubator. Alternatively, E-cadherin:DsRed MDCK cells were fixed in ice-cold methanol for 5 min, blocked in buffer containing 1% BSA, 1% GS and 1% DS and incubated with LGN antibody and subsequently Alexa-labelled secondary antibody. MDCK GFP-tubulin cells were imaged overnight (5 min per frame) on a customized Zeiss Observer inverted microscope (Intelligent Imaging Innovations, '3I') using a 20 × objective (N.A. 0.75) in a temperature and $CO_2$-controlled incubator. The orientation of the mitotic spindle in late metaphase (the frame before anaphase onset) and in anaphase was measured using ImageJ software (NIH). Binned data is shown in Fig. 3e, and unbinned data in Supplementary Fig. 3f. At late time points, a fraction of cells displayed extensive expansions at the wall with widths up to 200 μm. Furthermore, a minor fraction of cells associated with Fc-coated sidewalls divided perpendicular to the basal surface. Both of these were excluded from analyses.

**Imaging plasma membrane-associated astral microtubules.** MDCK cells expressing mRFP-EB1 were seeded onto glass-bottomed imaging dishes (Mattek) pre-coated with either poly-D-lysine (Sigma, according to manufacturer's instructions) or E-cadherin:Fc (functionalized similarly as sidewalls, see above). Six hours after seeding cells in metaphase with the mitotic spindle oriented perpendicular to the basal surface were live-imaged by Total Internal Reflection Fluorescence (TIRF) on a Zeiss Observer inverted microscope (3I) through a 100x Plan Fluor objective (1.45 N.A) and a 561 nm solid-state laser (CristaLaser) in a temperature and $CO_2$-controlled incubator. Images were taken at 1 s intervals for 60 s, and the dwell time of individual EB1 puncta within the TIRF plane (that were present for at least two consecutive frames) was measured using ImageJ software.

**Surface micropatterning of alternating protein stripes.** Micropatterning of alternating protein stripes was adapted from ref. 31 with some modifications. An array of 10 μm wide lines in a 10 μm pitch was laser-etched into a chrome mask (Photoplot International). Silicon wafers were then spin-coated with SU-8 2010 to a thickness of 10 μm, and exposed to 365 nm ultraviolet radiation through the photomask on a mask aligner (Karl-Suss). Patterns were subsequently developed in SU-8 developer and silanized with chlorotrimethylsilane. Polydimethylsiloxane (PDMS; Sylgard 182, Dow Corning) was cast onto the etched resist-coated wafer, cooked overnight at 60 °C, and peeled off the wafer. Stamps were oxidized with ultraviolet-ozone plasma (45 s, 50 W, 500 mTorr, PDC-001, Harrick Plasma) and inked with a solution of $160 \, \mu g \, ml^{-1}$ collagen-IV (Sigma) that was labelled with either cy3.5 or cy5.5 dye (Amersham) according to manufacturer's protocol. After incubation at room temperature for 20 min, stamps were dried under clear air flow and subsequently applied to glass-bottomed imaging dishes (Mattek) under load for 30 s and then gently peeled off after 2 min. Stamped imaging dishes were incubated with $50 \, \mu g \, ml^{-1}$ E-cadherin:Fc or human IgG Fc (Abcam) for 1 h and washed with PBS before seeding of $5 \times 10^4$ cells in each dish. TIRF microscopy of GFP-NuMA expressing cells was performed 3 h after plating using a Zeiss Observer inverted microscope (Intelligent Imaging Innovations, '3I') and a 100 × Plan Fluor (1.45 N.A.) objective (Olympus) and a 473 nm solid-state laser (CristaLaser) in a temperature and $CO_2$-controlled incubator. Alternatively, cells were fixed in ice-cold methanol for 5 min, blocked in buffer containing 1% BSA, 1% GS and 1% DS and incubated with LGN or NuMA antibody and subsequently Alexa-labelled secondary antibody followed by imaging by TIRF microscopy (see above). The fluorescence intensity of LGN/NuMA localized to stripes of E-cadherin:Fc, human IgG Fc or collagen-IV was measured with ImageJ (NIH), and local background fluorescence was subtracted. Statistical analysis of cells was performed using an unpaired student *t*-test.

**Mitochondrial targeting.** U2OS cells were co-transfected with YFP-tagged TPR repeats of human LGN (NM_013296.4, amino acids 8–351, cloned into a pcDNA3.1 citrine plasmid using Gateway cloning, Invitrogen). YFP–LGN–TPR or GFP-p120-catenin were co-expressed with the RFP-tagged mitochondrial localization signal of ActA[51], fused to either full cytosolic tail or JMD of E-cadherin (NM_009864.2, amino acids 736–888 and 736–783)[51], containing the indicated point mutations. Live-cell imaging was performed in medium contain 10 mM HEPES on a heat-controlled stage on an Axioskop2 LSM510 confocal microscope (Zeiss) with a 63 × objective (N.A. 1.4). The ratio of the amount of fluorescence of YFP–LGN–TRP or GFP-p120-catenin at the mitochondria relative to the total amount of fluorescence was measured in ImageJ (NIH). Statistical analysis was performed using an unpaired *t*-test with Welch correction.

**Data availability.** The authors declare that the data supporting the findings of this study are available within the paper and its Supplementary Information files.

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

## Acknowledgements

This work was supported by a NWO Rubicon fellowship (M.G.), KWF fundamental cancer research fellowship (M.G.), NIH CMB Training Grant (J.M.B.), NSF GRFP (J.M.B.), HHMI Life Science Research Foundation Fellowship (D.J.C.) and the NIH (GM35527, W.J.N.).

## Author contributions

M.G., J.M.B., K.A.S. and D.J.C. performed experiments, and analysed and interpreted data; M.G. and W.J.N. designed the study and experiments, and wrote the manuscript.

## Additional information

**Competing financial interests:** The authors declare no competing financial interests.

