## [Peer Review File · Nature Communications]

Reviewers' Comments:

Reviewer #2 (Remarks to the Author)

The authors have addressed most of the major issues that were raised in the initial review of this manuscript, and have added important new experimental data to bolster their conclusion that E-cadherin can function as an instructive cue for spindle orientation, by direct association with LGN. The competition experiment in Supp. Figure 6, and Galphai depletion experiment are particularly helpful. Overall, I think that this is an important contribution to the field.

I really have only a few minor points:

1. The knockdown data in Supplementary Figure 1 are not meaningful in isolation, since the point is to compare the staining in the KD cells with that in the control, under identical conditions of exposure time, antibody concentration, etc. Either this panel should be included as part of Figure 1a or an equivalent image to that shown in 1a (top) should be included in the Supplementary Figure 1.
2. Understandably the authors want to broaden the appeal of the study, but disruption of spindle orientation has not yet been identified as an oncogenic event, contrary to the statement in the Discussion. The cited review discusses the fact that loss of certain tumor suppressor proteins can cause spindle orientation defects but there is no evidence to my knowledge that such defects are causally related to cancer (as, indeed, is acknowledged in the review).
3. In the experiments using the T151 mutant of E-cadherin, are the tight junctions intact? (since loss of TJs could disrupt polarity and indirectly cause spindle misorientation).
4. Since E-cadherin binds at the same site on LGN as NuMA one would expect that LGN would be in the open conformation at adherens junctions, and would be able to interact with neighboring Galphai subunits – which could explain why the LGN does not diffuse away from the junction area when E-cadherin is displaced by NuMA – a possibility shown in the model figure 5h but not discussed.
5. The scale bar is missing in Supplementary Figure 6a.

Reviewer #3 (Remarks to the Author)

The authors have adequately addressed the reviewers comment from the initial round of review.

Reviewer #4 (Remarks to the Author)

I have been asked to comment specifically on the concerns of reviewer 1 and whether they have been addressed. The reviewer's main points revolve around the concern that "The stabilization of MTs [by E-cadherin] appears indirect". The authors responded by explaining their model in which (1) E-cadherin localizes LGN to cell-cell contacts, (2) NuMA displaces LGN from E-cadherin during mitosis, (3) the released NuMa-LGN complexes bind cortical G-alpha-i locally, and (4) the NuMa-LGN-G-alpha-i complexes stabilize MT plus ends in proximity to E-cadherin. In my opinion, the data strongly support steps 1-3. However, the test of step 4 is incomplete using their system. Only E-cadherin and LGN were tested for their roles stabilizing MT plus ends. The authors' model hinges on E-cadherin supplying LGN locally for formation of separate complexes cortically-anchored by G-alpha-i and MT-bound by NuMa. However, it was not shown whether G-alpha-i RNAi or NuMa RNAi reduce MT interactions with the artificial E-cadherin contacts to the same degree as LGN RNAi (data that should be added to Figure 6d)? These data would provide strong evidence for the final

step of the model using their elegant experimental system, and I think definition of the overall mechanism is of fundamental importance to our understanding of cell division in epithelial tissues (and alleviates the concern about an indirect role of E-cadherin). Without these two straightforward RNAi experiments, the story seems incomplete and thus the mechanism remains questionable.

I also agree with the reviewer, that the title is misleading. "Cell division orientation is directly coupled to cell-cell adhesion by the E-cadherin/LGN complex" could be changed to "Cell division orientation is coupled to cell-cell adhesion by the E-cadherin/LGN complex" (deleting "directly").

Similarly, the abstract is vague, and my first impression is a direct role for E-cadherin in anchoring MTs. It leaves out the direct role NuMa-LGN-G-alpha-i complex in MT anchoring. Specifically, "We show that LGN, which adopts a three dimensional structure similar to cadherin-bound catenins, binds directly to the E-cadherin cytosolic tail and localizes at cell-cell adhesions in interphase. This complex guides the mitotic recruitment of NuMA and stabilizes cortical associations of astral microtubules at cell-cell contacts to orient the mitotic spindle."

Similarly, I find the first paragraph of the discussion section technically correct but misleading, "Moreover, we identified the underlying molecular mechanism by which E-cadherin orients the mitotic spindle, through the successive, cell-cycle dependent recruitment of LGN and NuMA to E-cadherin based cell-cell contacts. This directs the formation of stable cortical attachments of mitotic microtubules to E-cadherin adhesions, and the alignment of the mitotic spindle with cell-cell contacts."

The reviewer's third concern about the use of Ric8 as a perturbation approach has been addressed effectively by employing an independent perturbation. This added G-alpha-i RNAi experiment could be added to Fig 5g instead of being supplemental.

Below, we have added the verbatim comments of the reviewers, and our responses to each in **bold**.

Reviewer #2 (Remarks to the Author):

The authors have addressed most of the major issues that were raised in the initial review of this manuscript, and have added important new experimental data to bolster their conclusion that E-cadherin can function as an instructive cue for spindle orientation, by direct association with LGN. The competition experiment in Supp. Figure 6, and Galphai depletion experiment are particularly helpful. Overall, I think that this is an important contribution to the field. I really have only a few minor points:

1. The knockdown data in Supplementary Figure 1 are not meaningful in isolation, since the point is to compare the staining in the KD cells with that in the control, under identical conditions of exposure time, antibody concentration, etc. Either this panel should be included as part of Figure 1a or an equivalent image to that shown in 1a (top) should be included in the Supplementary Figure 1.

Response: As requested, we have included the requested control to Supplementary Figure 1. We added a LGN stained image from the scr shRNA infected cells that was from the same experiment as the LGN shRNA infected cells.

2. Understandably the authors want to broaden the appeal of the study, but disruption of spindle orientation has not yet been identified as an oncogenic event, contrary to the statement in the Discussion. The cited review discusses the fact that loss of certain tumor suppressor proteins can cause spindle orientation defects but there is no evidence to my knowledge that such defects are causally related to cancer (as, indeed, is acknowledged in the review).

Response: We cited a publication that reported evidence for a causal role of spindle misorientation in tumorigenesis (Nakajima et al., Nature 2013); however, we only cited this article in the introduction, as background. Nonetheless, we agree with the reviewer that experimental evidence for a causal link between spindle misorientation and tumor formation is limited, and therefore we have weakened our statement on this in the Introduction, and removed a comment on this from the discussion.

In the introduction:

“The importance of correct division orientation is underlined by various developmental disorders that are a consequence of misoriented cell division^{5,6}, which may also contribute to tumor progression⁷⁻¹⁰.”

3. In the experiments using the T151 mutant of E-cadherin, are the tight junctions intact? (since loss of TJs could disrupt polarity and indirectly cause spindle misorientation).

Response: In the original publication, we showed that tight junctions are intact and functional in T151 E-cadherin expressing MDCK cells (Troxell et al., JCS 2000). We have stated this in the legend to Supplementary Figure 3a.

4. Since E-cadherin binds at the same site on LGN as NuMA one would expect that LGN would be in the open conformation at adherens junctions, and would be able to interact with neighboring Galphai subunits – which could explain why the LGN does not diffuse away from

the junction area when E-cadherin is displaced by NuMA – a possibility shown in the model figure 5h but not discussed.

Response: The hypothesis that $G\alpha_i$ may retain the LGN/NuMA complex at cell-cell adhesions when E-cadherin is displaced by NuMA has been the basis for our experiments with $G\alpha_i$ in Figure 5, and is discussed in the result section for this Figure (see excerpt below) as well as in the discussion.

“Following junctional enrichment of LGN by E-cadherin in interphase, the mitotic release of NuMA from the nucleus competes LGN from E-cadherin, as we demonstrated in vitro (Fig. 5a); the resulting LGN/NuMA complex remains localized at E-cadherin - based cell-cell contacts by LGN binding to cortical $G\alpha_i$ ¹¹. To test this hypothesis, we examined whether the recruitment of NuMA to E-cadherin adhesions was dependent on both LGN and the interaction of LGN with $G\alpha_i$. As we hypothesized, NuMA was not recruited to E-cadherin adhesions in mitosis upon shRNA-mediated depletion of LGN (Figs. 4g, S5a and S5b). Because LGN binding depends on the GDP-bound state of $G\alpha_i$ ¹¹, we next over-expressed the guanine nucleotide exchange factor Ric8 to drive $G\alpha_i$ into the GTP-bound state and thereby disrupt binding to LGN, as previously demonstrated³⁸. Similar to depletion of LGN, this resulted in loss of NuMA recruitment to micro-patterned E-cadherin adhesions in mitotic cells (Figs. 4g and S5c). Importantly, Ric8 overexpression did not affect the junctional localization of E-cadherin itself (Supplementary Fig. 5d). Finally, siRNA-mediated depletion of $G\alpha_i$ from MDCK cells resulted in a similar loss of NuMA recruitment to micro-patterned E-cadherin adhesions in mitosis (Fig. 5g).”

5. The scale bar is missing in Supplementary Figure 6a.

Response: The scale bars have been added

Reviewer #3 (Remarks to the Author):

The authors have adequately addressed the reviewers comment from the initial round of review.

Reviewer #4 (Remarks to the Author):

I have been asked to comment specifically on the concerns of reviewer 1 and whether they have been addressed. The reviewer’s main points revolve around the concern that “The stabilization of MTs [by E-cadherin] appears indirect”. The authors responded by explaining their model in which (1) E-cadherin localizes LGN to cell-cell contacts, (2) NuMA displaces LGN from E-cadherin during mitosis, (3) the released NuMa-LGN complexes bind cortical G-alpha-i locally, and (4) the NuMa-LGN-G-alpha-i complexes stabilize MT plus ends in proximity to E-cadherin. In my opinion, the data strongly support steps 1-3. However, the test of step 4 is incomplete using their system. Only E-cadherin and LGN were tested for their roles stabilizing MT plus ends. The authors’ model hinges on E-cadherin supplying LGN locally for formation of separate complexes cortically-anchored by G-alpha-i and MT-bound by NuMa. However, it was not shown whether G-alpha-i RNAi or NuMa RNAi reduce MT interactions with the artificial E-cadherin contacts to the same degree as LGN RNAi (data that should be added to Figure 6d)? These data would provide strong evidence for the final step of the model using their elegant experimental system, and I think definition of the overall mechanism is of fundamental importance to our understanding of cell division in epithelial tissues (and alleviates the concern

about an indirect role of E-cadherin). Without these two straightforward RNAi experiments, the story seems incomplete and thus the mechanism remains questionable.

Response: We realize we may not have emphasized adequately the extensive published data on the link between LGN and NuMA. It is very well established by different laboratories that LGN orients the mitotic spindle in mammalian cells through NuMA; NuMA is critical as it binds to astral microtubules and the microtubule motor protein dynein. This is widely accepted in the field, as for instance indicated in all reviews on this topic (see for instance *Di Pietro et al., Embo Reports 2016*). Moreover, this point was emphasized by Reviewer #1, who noted “The role of LGN and NuMA in spindle orientation/positioning is pretty clear”. As further indicated by Reviewer #1, and by ourselves in the original rebuttal letter to the first round of revisions, the key advance of our findings the demonstration that E-cadherin orients the mitotic spindle by mediating the cell-cycle dependent recruitment of LGN, and then NuMA which competes LGN from E-cadherin resulting in local retention of the LGN/NuMA complex by $G\alpha_i$ at cell-cell contacts.

Reviewer #1 wanted us to avoid implying that there is a direct connection between E-cadherin and astral microtubules. He had not asked us for additional experiments to show that E-cadherin and LGN orient the spindle through NuMA, because, as indicated above, it has already been well established that NuMA regulates the association of astral microtubules with the cortex and thereby spindle orientation downstream of LGN. Importantly, we have clearly demonstrated that LGN binds directly to E-cadherin, that NuMA is recruited to cell-cell adhesions by E-cadherin/LGN, and that this recruitment of NuMA requires $G\alpha_i$. We acknowledge that we needed to strengthen the text and provide more background on previous publications on the link between LGN and NuMA. However, we strongly believe that our model is sufficiently supported by our current experiments together with existing data, and that Reviewer #1 did not question the requirement of NuMA downstream of LGN in our experiments.

The discussion of Figure 6 in the text now starts with:

“NuMA provides the link between LGN and the mitotic spindle, as NuMA binds directly to astral microtubules and is able to apply a pulling force on those microtubules through its association with dynein^{11,13-16}. Having established that the E-cadherin/LGN complex directs the mitotic localization of NuMA to cell-cell adhesions (Figure 5), we hypothesized that mitotic microtubules should form more stable associations with the cell cortex at E-cadherin-dependent cell-cell adhesions than at non-adhesion sites.”

I also agree with the reviewer, that the title is misleading. “Cell division orientation is directly coupled to cell-cell adhesion by the E-cadherin/LGN complex” could be changed to “Cell division orientation is coupled to cell-cell adhesion by the E-cadherin/LGN complex” (deleting “directly”).

Response: As requested we have removed “directly” from the title

Similarly, the abstract is vague, and my first impression is a direct role for E-cadherin in anchoring MTs. It leaves out the direct role NuMa-LGN-G-alpha-i complex in MT anchoring. Specifically, “We show that LGN, which adopts a three dimensional structure similar to cadherin-bound catenins, binds directly to the E-cadherin cytosolic tail and localizes at cell-cell adhesions in interphase. This complex guides the mitotic recruitment of NuMA and stabilizes cortical associations of astral microtubules at cell-cell contacts to orient the mitotic spindle.”

Response: We have altered the Abstract to more clearly reflect our data on the mechanism by which E-cadherin regulates LGN/NuMA localization and, thereby, astral microtubule anchoring, and to avoid any impression that E-cadherin itself is directly coupled to astral microtubules.

“We show that LGN, which adopts a three-dimensional structure similar to cadherin-bound catenins, binds directly to the E-cadherin cytosolic tail and localizes at cell-cell adhesions in interphase. Upon mitotic entry, NuMA is released from the nucleus and competes LGN from E-cadherin to locally form the LGN/NuMA complex. The LGN/NuMA complex mediates the stabilization of cortical associations of astral microtubules at cell-cell adhesions to orient the mitotic spindle. These results show how E-cadherin instructs the assembly of the LGN/NuMA complex at cell-cell contacts, and define a mechanism that couples cell division orientation to intercellular adhesion.”

Similarly, I find the first paragraph of the discussion section technically correct but misleading, “Moreover, we identified the underlying molecular mechanism by which E-cadherin orients the mitotic spindle, through the successive, cell-cycle dependent recruitment of LGN and NuMA to E-cadherin based cell-cell contacts. This directs the formation of stable cortical attachments of mitotic microtubules to E-cadherin adhesions, and the alignment of the mitotic spindle with cell-cell contacts.”

Response: We have altered the text to avoid the impression that NuMA is directly bound to E-cadherin:

“Moreover, we identified the underlying molecular mechanism by which E-cadherin orients the mitotic spindle. E-cadherin directly mediates the recruitment of LGN to cell-cell adhesions, and upon mitotic entry LGN is competed from E-cadherin by NuMA, resulting in local formation of the LGN/NuMA complex that is retained at cell-cell contacts by $G\alpha_i$. This directs the formation of stable cortical attachments of mitotic microtubules to E-cadherin-based adhesions”

The reviewer’s third concern about the use of Ric8 as a perturbation approach has been addressed effectively by employing an independent perturbation. This added G-alpha-i RNAi experiment could be added to Fig 5g instead of being supplemental.

Response: As suggested, we have included this experiment in Figure 5g.